# Incidence and Severity of Community- and Hospital-Acquired Hyponatremia in Pediatrics

**DOI:** 10.3390/jcm11247522

**Published:** 2022-12-19

**Authors:** J. M. Rius-Peris, P. Tambe, M. Chilet Sáez, M. Requena, E. Prada, J. Mateo

**Affiliations:** 1Pediatric Department, Virgen de la Luz Hospital, 16002 Cuenca, Spain; 2Medical Analysis Expert Group, Institute of Technology, Universidad de Castilla-La Mancha, 16071 Cuenca, Spain; 3Pediatric Department, Southland Hospital, Invercargill 9812, New Zealand; 4Analysis and Microbiology Department, Virgen de la Luz Hospital, 16002 Cuenca, Spain; 5Computer Analysis Department, Virgen de la Luz Hospital, 16002 Cuenca, Spain; 6Clinical Analysis Department, Virgen de la Luz Hospital, 16002 Cuenca, Spain

**Keywords:** community-acquired hyponatremia, hospital-acquired hyponatremia, incidence, related factors

## Abstract

Hyponatremia is the most common electrolyte disturbance in hospitalized children, with a reported incidence of 15–30%, but its overall incidence and severity are not well known. The objective of our study was to determine the incidence, severity, and associated risk factors of community- and hospital-acquired hyponatremia on a general pediatric ward. Data of 5550 children admitted from June 2012 to December 2019 on plasma sodium and discharge diagnosis were analyzed by logistic regression model. Clinically relevant diagnostic groups were created. Hyponatremia was classified as mild, moderate, and severe. The incidence of community- and hospital-acquired hyponatremia was 15.8% and 1.4%, respectively. Most of the cases were mild (90.8%) to moderate (8.6%), with only two cases of severe community-acquired hyponatremia. There were no clinical complications in any of the hyponatremic children. Age and diagnosis at discharge were principal factors significantly correlated with hyponatremia. Community-acquired hyponatremia is more common than hospital-acquired hyponatremia in clinical practice. Severe cases of both types are rare. Children from 2 to 11 years of age presenting with infections, cardiovascular disorders, and gastrointestinal disorders are at risk of developing hyponatremia.

## 1. Introduction

Hyponatremia is defined as when a plasma sodium level falls below 135 mEq/L [1]. It is believed to be the most common electrolyte disturbance, with an estimated incidence of 15 to 30% in hospitalized children [1]. Not only the disorder itself but also its treatment has been associated with increased morbidity and mortality in adults and children [2,3]. Plasma sodium is maintained by the interplay of antidiuretic hormone (ADH), renin–angiotensin–aldosterone axis and natriuretic peptides. Hyponatremia results mostly from retention of free water because of increased levels of ADH caused by osmotic or non-osmotic stimuli. Acute hyponatremia results in an influx of free water into the intracellular space, leading to cerebral oedema and noncardiogenic pulmonary oedema [4]. This can result in permanent neurological injury and death, especially when plasma sodium falls below 125 mEq/L [1]. Postoperative hyponatremia is also a serious problem and deaths are reported in the scientific literature because of hyponatremic encephalopathy following routine surgical procedures [5]. There are limited studies on the overall incidence of hyponatremia in children, with wide variation in the reported incidence rates in hospitalized children. The heterogeneity in the scientific literature is due to some factors, such as cut-off values of plasma sodium level [6], scope of patient care (general ward or intensive care) [1], process of hospitalization (emergency, elective, medical, or surgical) [7,8], and studies limited to specific clinical pathologies, such as gastroenteritis [9], bronchiolitis [9], pneumonia [10], meningitis [11], pyelonephritis [9], and others [12,13]. Therefore, it is difficult to validate the incidence of hyponatremia in hospitalized children on a general pediatric ward from published data.

Some adult studies have differentiated community-acquired hyponatremia (CAH), hyponatremia present at admission, and hospital-acquired hyponatremia (HAH), hyponatremia occurring at some point during hospitalization with normal levels of sodium on admission [6]. We consider these concepts as an incipience due to dearth of such studies in the pediatric population.

We hypothesized that CAH was a more common phenomenon for clinical practice in the general pediatric ward setting in comparison to HAH. The goals of this study were to estimate the incidence of CAH and HAH with its severity on the general pediatric ward and to explore some explanatory factors associated with this phenomenon.

## 2. Materials and Methods

### 2.1. Study Design and Setting

An observational and retrospective study of a cohort of patients admitted to the pediatric general ward at a secondary care University Hospital from 19 June 2012 to 31 December 2019, in Cuenca, Castilla La Mancha (Spain), was undertaken. The general pediatric ward admits medical patients from 28 days of life to 18 years of age and general surgical patients from 3 years to 18 years of age.

### 2.2. Sample, Inclusion, and Exclusion Criteria

All medical and surgical patients, independent of the type of admission (emergency/planned/referred), who had at least one plasma sodium measured were included in the study. Children with malignancies (140-239; C00-D48) are managed at a tertiary center and, hence, were not included in the study sample. No other exclusion criteria were used.

### 2.3. Data Collection

Data to be analyzed for each patient were obtained from the Data Warehouse and Data Exploitation Platform (DW & DEP). This platform incorporates its information through data extraction, transformation, and load processes (ETL) of the different information systems.

The information uploaded to the DW & DEP was extracted from the Hospital Information System (HIS), which included the main and secondary diagnosis at discharge and all procedures coded by the International Classification of Diseases (ICD-9 and 10). Laboratory data were included through integration between HIS and Laboratory Information System (LIS). This information was provided as three separate tables on hospitalization (admission details), coding (ICD codes), and laboratory data.

### 2.4. Data Cleaning

The three tables provided by the DW & DEP were subsequently exploited with the required criteria and crossed by common fields related to each other. This resulted in a final table in which each record was equivalent to a single episode of admission.

### 2.5. Data Mining

Data for children with at least one plasma sodium level were reviewed for main and secondary diagnosis at discharge. This was coded according to International Classification of Diseases, Ninth Revision (ICD-9), from June 2012 to December 2015, and Tenth Revision (ICD-10), from January 2016 to December 2019. PT and JMR reviewed the main and secondary discharge diagnosis for each patient and appropriate reallocation of the main and secondary diagnosis was performed (e.g., cases recorded with the main diagnosis of chronic tonsillitis and adenoiditis (474, J35) and secondary diagnosis of adenotonsillectomy (28.2, 0CTPXZZ), the main diagnosis was changed from respiratory disorders to surgical). Any discrepancies were resolved with discussion and consensus reached. The ICD codes were then utilized to categorize cases into clinically relevant diagnostic groups, an approach which has been used before [14]. All infections were identified and grouped together irrespective of the system involved. Symptoms, signs, and abnormal clinical and laboratory findings not elsewhere classified (780-799, R0-99) were evaluated and reassigned to the most appropriate clinical discharge diagnosis. Admissions for poisoning and allergic disorders were separated from the group of external causes of morbidity and mortality (960-979, T36-78). Surgical category included all cases that needed surgical intervention as per the procedural ICD codes. The diagnostic group Trauma included admissions that did not need any surgical intervention. Kawasaki’s disease (446.1; M30.3) was included in the cardiovascular disorders group in view of the follow-up with cardiologist. Appendix A provides details on various diagnoses, with their ICD 9 and ICD 10 codes in the respective diagnostic groups.

### 2.6. Variables

#### 2.6.1. Hyponatremia

The primary outcome was presence of hyponatremia in our sample. The sodium ion was determined using the indirect ion-selective electrode (ISE) technique on the Cobas c501 analyzer (Serial No. 0829-19) (Roche Diagnostics). Precise data on date and time of admission, discharge, and time of sodium level determination were available for each patient. All sodium levels of each patient from 12 h before admission (from emergency area) until the end of the hospitalization process were retrieved. Plasma sodium level of the first blood sample obtained during the care process (emergency and elective admissions) of each patient was considered for the incidence of CAH. Any further low sodium level/s after the first normal one was considered for the incidence of HAH. The lowest sodium level measured during the care process was considered for the incidence of the phenomenon under study. Hyponatremia was considered when sodium concentration was <135 mEq/L and was classified as mild (130–134 mEq/L), moderate (125–129 mEq/L), and severe (<125 mEq/L) [15].

Nursing staff follow standard protocol on our pediatric ward for blood sample collection from peripheral veins rather than the catheter site to avoid pseudohyponatremia. The analytical system used in our laboratory for plasma sodium evaluates the quality in all the samples, through the automated determination of the known serum indices (hemolysis, lipemia, and jaundice) that affect sodium level. The system then cancels the processing of these samples that would be altered by the value of any of the indices, including lipemia [16].

#### 2.6.2. Age

Children admitted from 1 month to 18 years of age were included in the study. The study population was stratified according to an accepted international consensus [17].

#### 2.6.3. Diagnostic Group (DG)

The diagnostic groups, as described in data mining, were considered as a secondary variable.

### 2.7. Analysis

Qualitative variables were summarized as absolute frequencies and percentages and compared by the χ^2^ test. Normally distributed quantitative variables were summarized as mean and standard deviation (SD) and compared by the Student t test. Quantitative variables, such as age, which did not follow a normal distribution were summarized as median and interquartile range (IQR) and Mann–Whitney U test was used for the comparison. A bivariate analysis was performed first for CAH as a primary outcome. Subsequently, multivariable analysis by means of stepwise binary logistic regression model (forward: LR method for including variables) was performed. The strength and direction of the association were reported with odds ratios (ORs) and 95% confidence intervals (CIs). Associations were considered clinically relevant when they were statistically significant (*p* < 0.05) and showed high effect size in categorical variables (OR: <0.9 or >1.1). The statistical analysis was performed with the SPSS software for Windows, version 21.0 (SPSS Inc., Chicago, IL, USA).

### 2.8. Ethical Considerations

The study protocol was approved by the Clinical Research Ethics Committee of the Hospital on 24 April 2019 (registered number: 2019/PI0619). We adhered to the principles of the Declaration of Helsinki and the regulatory standards established in Organic Law 3/2018 of 5 December for the Protection of Personal Data and Guarantee of Digital Rights. Pseudonymized data were made available by the team, which was technically and functionally separate from the research team, after commitment to confidentiality and nonindulgence in reidentification. Specific security measures were adopted to prevent reidentification and access by unauthorized third parties.

## 3. Results

### 3.1. Demographics

During the study period, 5550 patients were admitted and 3102 of them had at least one blood sample for plasma sodium determination, which constituted our sample for analysis. Median age of patients was 7.43 years (IQR: 2.49–13.09) and the mean length of stay (LOS) was 2.76 ± 3.19 days. Infections 1349/3102 (43.5%), surgical 598/3102 (19.3%), and gastrointestinal disorders 308/3102 (9.9%) were the most common diagnostic groups (Table 1).

### 3.2. Incidence and Severity of Hyponatremia

During the study period, there were 492 cases of CAH and 42 of HAH, with an incidence of 15.8% and 1.4%, respectively. Most of the cases had mild 486/534 (91.0%) to moderate 46/534 (8.6%) hyponatremia. There were only two cases with severe hyponatremia 2/534 (0.4%), which were CAH. The severe cases were from the infection group, one each with acute pyelonephritis and acute gastroenteritis. There were no complications, such as encephalopathy, pulmonary oedema, and deaths, reported due to hyponatremia during the study period. The incidence of CAH and HAH in relation to age groups and severity is presented (Table 2), as well as in relation to diagnostic groups and severity (Table 3).

### 3.3. Associated Factors with Hyponatremia

Gender did not contribute to bivariate analysis. Patients admitted in summer present lower incidence of hyponatremia in comparison to those admitted during the rest of the year. Bivariate and multivariate analysis of the data indicate an association between age and diagnostic groups with CAH. The strongest association was with infections (OR: 6.88; 95% CI 2.76–17.16) and cardiovascular disorders (OR: 4.66; CI 1.41–15.35) (Table 4).

### 3.4. Groups with Special Considerations on Hyponatremia

Children with diabetic ketoacidosis were noted to be pseudohyponatremic because of hyperglycemia [18]. Thus, only one infant with diabetic ketoacidosis had factual moderate hyponatremia. Hyponatremia in the Pregnancy and Childbirth category is defined as plasma sodium levels of <130 mEq/L [19]. Hence, there were no cases of hyponatremia in this adolescent group.

### 3.5. Subgroup Analysis of Diagnostic Group Infections

A subgroup analysis of the infections according to the anatomical site was conducted. This is depicted in Table 5, suggesting septicemia (62.5%), diarrhea (28.7%), and pneumonia (27.8%) were the commonest infections associated with CAH.

### 3.6. Progression of Sodium Levels within the Sample

Information about the progression of sodium levels of the patients with CAH and HAH is shown in Figure 1 and Figure 2.

## 4. Discussion

The pediatric studies from which evidence on the incidence of hyponatremia is based are heterogeneous in terms of inclusion criteria and conceptualization of the phenomenon under study. There is variation in the cut-off point for hyponatremia between the studies and they are not classified according to its severity [9,10,11]. Most of them were not designed under the main double objective of describing incidence and severity of this phenomenon in a wide variety of diagnoses from a sufficiently large pediatric sample. This would reduce the reliability of the outcome. Hence, our study involved an analysis of hyponatremia in a diverse hospitalized pediatric population of all clinical diagnostic groups.

Incidence of CAH and HAH has been reported as 14.3% and 8.1% by Hoorn et al. [20], while Al Shibli et al. [21] reported an incidence of 36% and 9%, respectively. This is incongruent from our reported HAH incidence of 1.4%. Both studies defined hyponatremia at or below the plasma sodium of 135 mEq/L. Inclusion of children with plasma sodium levels of 135 mEq/L could have contributed to the increase in the incidence of hyponatremia in these studies [20,21]. This can be shown in our data where incidence increased from 492/3102 (15.8%) to 772/3102 (24.9%) for CAH and from 42/3102 (1.4%) to 71/3102 (2.3%) for HAH. In addition, the study by al Shibli A et al. included children only with bronchiolitis [21].

In a study of children from 12 months to 18 years of age, the overall incidence of hyponatremia was reported as 1.38%, with 43% hyponatremic on admission and 57% with HAH. The authors described hyponatremia as plasma sodium <130 mEq/L, with mild (125–129 mEq/L), moderate (120–124 mEq/L), and severe (<120 mEq/L) hyponatremia, respectively. Hence, our study is not comparable due to the different inclusion criteria [22].

In a prospective study, 27.7% of children admitted to the pediatric ward and intensive care unit were reported with HAH. Further analysis on 46.8% of children who were hyponatremic on admission was not undertaken. The high incidence of HAH was attributed to use of hypotonic intravenous fluids in 96.7% of children. Children with HAH were categorized in various clinical disorders, the basis of this classification had not been described. There was no surgical category even though 37.5% of the children with HAH underwent surgery [23]. This makes direct comparison less relevant with our data.

It is evident from these studies that the incidence of HAH was much lower than CAH, which is especially clear from our data. Therefore, CAH is quantitatively more relevant than HAH in a secondary-level pediatric setting. In the studies that report hyponatremia classified by severity, the majority were mild forms [8,10,11,21,23,24], as is also a finding in our study.

We report higher incidence of CAH in children from 2 to 11 years and HAH more frequent from 5 to 17 years (Table 2). Other studies have reported higher incidence of hyponatremia in younger pediatric patients [8,9,25] but similar age group analysis has not been conducted. The highest incidences of CAH in our study were in infections, cardiovascular, and hematological and immune disorders groups (Table 3). In contrast, HAH was more frequent in allergy and toxins, endocrine and metabolic disorders, and gastrointestinal disorders (Table 3).

From the multivariable analysis, it can be concluded that the highest risk of CAH was in children included in diagnostic groups: infections, cardiovascular disorders, and gastrointestinal disorders. To our knowledge, a combined analysis of the variables age, period of the year, and diagnosis at discharge has not been reported in the literature.

The incidence of HAH in our research was not only low, but all cases were mild forms in contrast to severe cases of HAH reported in the literature [26]. In our pediatric general ward, before October 2014, all children who needed intravenous fluids received hypotonic fluids (sodium chloride (NaCl) 0.20% + dextrose (D) 5% or NaCl 0.33% + D 5%). Gradually, after October 2014, this was replaced with isotonic fluids (NaCl 0.9% + D 5%) [27]. However, there was no difference in incidence of HAH before and after October 2014 (Table 6). In a retrospective study of the change in maintenance fluids from hypotonic to isotonic on the pediatric surgical ward, the prevalence of hyponatremia decreased from 29% to 22% for all pediatric age groups (adjusted OR 0.61, CI 0.51–0.82). However, it is interesting to note that the prevalence of hyponatremia was still 22%, even with isotonic fluids [28], in contrast to that of HAH in our analysis and the literature reports of HAH [1]. Exclusion of surgical children under 3 years, children with complex surgeries, and transfer of critically ill children to tertiary intensive care unit after initial stabilization may be contributory factors towards the low incidence of HAH.

Infections contributed to 27.8% of children with CAH in our study, with septicemia (62.5%), pyelonephritis (32.4%), and pneumonia (27.8%) commonly associated with hyponatremia, which is similar to other studies [9,10,25,29]. Hyponatremia is well documented in children with febrile disease and infections. In a retrospective 5-year study, a significantly higher proportion of children with fever and febrile convulsion had hyponatremia, which was attributed to excess ADH secretion [30]. Liamis et al. reviewed the pathogenesis of hyponatremia in infections, which is categorized as hypertonic hyponatremia (infection-induced hyperglycemia), isotonic hyponatremia (infection-induced hyperproteinemia), and (hypervolemic, euvolemic, or hypovolemic) hypotonic hyponatremia [31]. Most of the infections cause euvolemic hyponatremia due to SIADH and secondary adrenal insufficiency but it is considered to be multifactorial in origin. Inflammatory cytokines, such as interleukin (IL)-1β and IL-6, have been shown to induce activation of ADH, resulting in hyponatremia [32]. It was proposed that the degree of inflammation reflected as raised white blood count (WBC) and increased C-reactive protein (CRP), rather than the site of inflammation, might be involved in the development of hyponatremia [33]. Further studies are needed to confirm this, which would be a clinically useful indicator. Our study was limited in this aspect as we could not associate the severity of infection either clinically or biochemically to the development of hyponatremia.

Various studies have reported CAH in about 30% to 45% [10,29] of children, with pneumonia due to SIADH, volume depletion, increased levels of IL-6, and renal impairment as possible mechanisms [31]. This is similar to our study, with 26.8% and 27.8% of children with upper respiratory tract infections (URTI) and pneumonia, respectively, with hyponatremia.

About 32% of children with pyelonephritis had mild CAH in our study, which is lower than that reported (67%) in a study by Milani et al. [25]. This may be due to inclusion of only children from 4 weeks to 24 months of age, as compared a wide range of age group in our study. There were only 20.6% of children with mild CAH due to meningitis in our investigation, which is similar to the study published by Lin et al., albeit there is wide variation in the incidence of CAH in meningitis of 50% to 66% in other studies [34]. The literature reports an incidence of about 18% to 36% of children [35] with gastroenteritis having CAH, which is similar to our study, with an incidence of 28.7%. Unfortunately, the present study could not ascertain treatment-based progression of any of these illnesses.

Our study has some limitations due to its single center and retrospective design. There was no provision to assess the relationship of hyponatremia with clinical severity scoring systems due to the data protection laws. Administration of intravenous fluids is recorded as a procedure code. Detailed data about type, rate, and duration of intravenous fluids are not recorded in the HIS. These details are recorded in the physical notes, prescription, and nursing documents, which were not accessible due to the national data protection laws.

We selected a representative sample of pediatric hospitalized patients by including an overly broad range of age, diagnosis, and method of admission (emergency or scheduled). About 60–70% of children in the most common diagnostic groups had their bloods analyzed for plasma sodium levels (Table 1). The study design is based on conceptual classification previously proposed by others [6]. In addition, the incidence and severity of hyponatremia was compared with diagnostic groups derived from the coding of discharge diagnosis. This approach was used previously for comparison of child mortality [14]. The quality (validity, reliability, completeness, and timeliness) of retrospective coded data depends on the documentation of discharge diagnosis and treatment by health care physicians and on accuracy and consistency of the coders. In our hospital, the health record coders are professionally trained through an accredited training program and supervised by a co-ordinator for consistent coding practice. The study by Hennessy et al. suggested that coder characteristics do not influence the validity of hospital discharge data, which can be achieved with appropriate training and management of the coding staff [36]. Therefore, we consider that our results are representative of a true incidence of the phenomenon at study.

To our knowledge this may be the first study with conceptual classification (CAH and HAH) of incidence and severity of hyponatremia in the hospitalized pediatric population. In addition, comparison with a broad set of ICD coded diagnoses grouped with a clinically and practically relevant classification provides valuable information on the incidence of hyponatremia in the pediatric population.

## 5. Conclusions

Overall incidence of hyponatremia in pediatric hospitalized children seems to be lower than that described in the literature and depends on factors related to the study population and the definition criteria for hyponatremia. Community-acquired hyponatremia (CAH) seems to be quantitatively more relevant than hospital-acquired hyponatremia (HAH) in clinical practice, especially in children from two to eleven years old admitted with infections, cardiovascular, and gastrointestinal disorders, a very important target/issue for pediatricians in primary and secondary hospital level settings. Further collaborative studies with unifying criteria for hyponatremia and its severity in the general pediatric population are needed. Thus, the true incidence of this electrolyte disturbance, classified by severity and diagnostic groups, will be known and the relevance of this event in clinical practice will be ascertained.

## Figures and Tables

**Figure 1 jcm-11-07522-f001:**
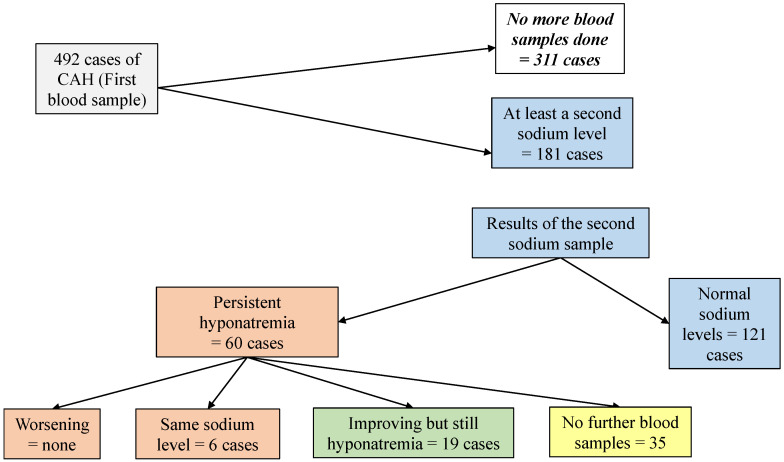
Progression of community-acquired hyponatremia along the admission.

**Figure 2 jcm-11-07522-f002:**
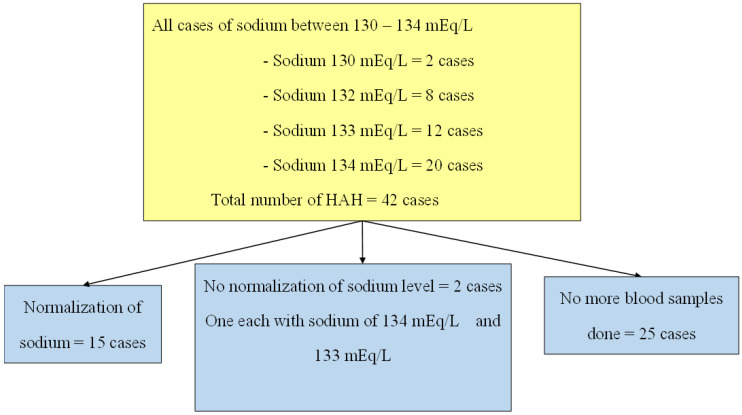
Progression of hospital-acquired hyponatremia along the admission.

**Table 1 jcm-11-07522-t001:** Baseline characteristics of the sample.

**Sex, male, n (%)**	1712 (55.2)	
**Median age (IQR), y**	7.43 (2.49–13.09)	
**Age group, n (%)**		
Infant, 0–1 y	326 (10.5)	
Toddler, 2–4 y	316 (10.2)	
Early childhood, 5–11 y	708 (22.8)	
Middle childhood, 12–14 y	804 (25.9)	
Early adolescent, 15–17 y	948 (30.6)	
Mean Length of stay (±SD), d	2.76 (3.19)	
**Diagnostic groups, n (%)**	**Total Cases**	**Cases with Blood Sample**
Infections	1930 (34.8)	1349 (43.5)
Surgical	1399 (25.2)	598 (19.3)
Gastrointestinal disorders	462 (8.3)	308 (9.9)
Neurological disorders	422 (7.6)	277 (8.9)
Respiratory disorders	302 (5.4)	95 (3.1)
Trauma	221 (4.0)	82 (2.6)
Endocrine and Metabolic disorders	182 (3.3)	71 (2.3)
Genito–urinary disorders	91 (1.6)	70 (2.3)
Allergy and Toxins	104 (1.9)	56 (1.8)
Mental Health disorders	72 (1.3)	49 (1.6)
Cardiovascular disorders	77 (1.4)	48 (1.5)
Miscellaneous	89 (1.6)	30 (1.0)
Hematological and Immune disorders	47 (0.8)	22 (0.7)
Musculoskeletal disorders	31 (0.6)	17 (0.5)
Eye disorders	28 (0.5)	12 (0.4)
Pregnancy and Childbirth	62 (1.1)	10 (0.3)
Skin and Subcutaneous disorders	31 (0.6)	8 (0.3)
Total	5550 (100)	3102 (100)

**Table 2 jcm-11-07522-t002:** Hyponatremia classified by kind, severity, and age.

Type of Hyponatremia	CAH				HAH ^a^	
Age Group/Severity of Hyponatremia	Mild	Moderate	Severe	Total	Mild	Total
All ages n, (%) 3102 ^b^	444 (90.2)	46 (9.3)	2 (0.4)	492 (15.8) ^c^	42	42 (1.4) ^d^
Infancy, 0–1 y 326 ^b^	52 (92.9)	4 (7.1)	-	56 (17.2) ^c^	1 (100)	1 (0.3) ^d^
Toddler, 2–4 y 316 ^b^	79 (88.8)	10 (11.2)	-	89 (28.2) ^c^	2 (100)	2 (0.6) ^d^
Early childhood, 5–11 y 708 ^b^	164 (88.2)	22 (11.8)	-	186 (26.3) ^c^	11 (100)	11 (1.6) ^d^
Middle childhood, 12–14 y 804 ^b^	97 (91.5)	8 (7.5)	1 (0.9)	106 (13.2) ^c^	12 (100)	12 (1.5) ^d^
Early adolescence, 15–17 y 948 ^b^	52 (94.5)	2 (3.6)	1 (1.8)	55 (5.8) ^c^	16 (100)	16 (1.7) ^d^

CAH, community-acquired hyponatremia; HAH, hospital-acquired hyponatremia. The hyponatremia incidences in relation to age group and grade of severity are expressed by absolute number and its percentage with respect to the total in parentheses. ^a^ All cases of HAH were mild. ^b^ Total cases and cases in each age group with at least one sodium level, expressed in absolute number. ^c^ The number in parentheses is the incidence of CAH out of the total cases in each age group. ^d^ The number in parentheses is the incidence of HAH out of the total cases in each age group.

**Table 3 jcm-11-07522-t003:** Hyponatremia classified by kind, severity, and diagnosis.

Type of Hyponatremia	CAH				HAH ^a^	
Diagnostic Group/Severity of Hyponatremia	Mild	Moderate	Severe	Total	Mild ^a^	Total
Total n, (%) 3102 ^b^	444 (90.2)	46 (9.3)	2 (0.4)	492 (15.8) ^c^	42 (100)	42 (1.4) ^d^
Infections 1349 ^b^	339 (90.6)	33 (8.8)	2 (0.5)	374 (27.7) ^c^	20 (100)	20 (1.5) ^d^
Surgical 598 ^b^	34 (94.4)	2 (5.6)	-	36 (6.0) ^c^	8 (100)	8 (1.3) ^d^
Gastrointestinal disorders 308 ^b^	34 (87.2)	5 (12.8)	-	39 (12.7) ^c^	8 (100)	8 (2.6) ^d^
Neurological disorders 277 ^b^	11 (84.6)	2 (15.4)	-	13 (4.7) ^c^	1 (100)	1 (0.4) ^d^
Respiratory disorders 95 ^b^	4 (80.0)	1 (20.0)	-	5 (5.3) ^c^	-	-
Trauma 82 ^b^	1 (100)	-	-	1 (1.2) ^c^	-	-
Endocrine & Metabolic disorders 71 ^b^	2 (66.7)	1 (33.3)	-	3 (4.2) ^c^	2 (100)	2 (2.8) ^d^
Genito-urinary disorders 70 ^b^	5 (83.3)	1 (16.7)	-	6 (8.6) ^c^	-	-
Allergy and toxins 56 ^b^	3 (100)	-	-	3 (5.4) ^c^	2 (100)	2 (3.6) ^d^
Mental Health disorders 49 ^b^	1 (100)	-	-	1 (2.0) ^c^	1 (100)	1 (2.0) ^d^
Cardiovascular disorders 48 ^b^	7 (87.5)	1 (12.5)	-	8 (16.7) ^c^	-	-
Hematological & Immune disorders 22 ^b^	3 (100)	-	-	3 (13.6) ^c^	-	-

CAH, community-acquired hyponatremia; HAH, hospital-acquired hyponatremia. The hyponatremia incidences in relation to diagnostic group and grade of severity are expressed by absolute number and its percentage with respect to the total in parentheses. Miscellaneous, musculoskeletal disorders, eye disorders, pregnancy and childbirth, skin and subcutaneous disorders are all diagnostics not presented in this table because no cases of hyponatremia were noted. ^a^ All cases of HAH were mild. ^b^ Total cases and cases in each age group with at least one sodium level, expressed in absolute number. ^c^ The number in parentheses is the incidence of CAH out of the total cases in each diagnostic group. ^d^ The number in parentheses is the incidence of HAH out of the total cases in each diagnostic group.

**Table 4 jcm-11-07522-t004:** Associated factors to community-acquired hyponatremia in pediatric patients (bivariate and multivariate analysis).

Total Cases, n (%)	Bivariate Analysis			Multivariate Analysis	
Secondary Variables	No Hyponatremia	Hyponatremia	*p-*Value	OR (CI)	*p-*Value
Sex			0.254	-	*-*
Female, 1390 (44.8)	1158 (83.3)	232 (16.7)			
Male, 1712 (55.2)	1452 (84.8)	260 (15.2)			
Age groups			<0.0001		
Infant, 326 (10.5)	270 (82.8)	56 (17.2)		1.64 (1.07–2.51)	0.022
Toddler, 316 (10.2)	227 (71.8)	89 (28.2)		3.19 (2.14–4.76)	<0.0001
Early child, 708 (22.8)	522 (73.7)	186 (26.3)		3.56 (2.51–5.05)	<0.0001
Middle child, 804 (25.9)	698 (86.8)	106 (13.2)		2.05 (1.44–2.93)	<0.0001
Early adolescent, 948 (36.6) ^a^	893 (94.2)	55 (5.8)		Reference	
Periods of the year			0.049	-	-
Winter, 786 (25.3)	655 (83.3)	131 (16.7)			
Summer, 585 (18.9)	511 (87.4)	74 (12.6)			
Rest of the year, 1731 (55.8)	1444 (83.4)	287 (16.6)			
Diagnostic groups			<0.0001		
Respiratory disorders, 95 (3.1) ^a^	90 (94.7)	5 (5.3)		Reference	
Infections, 1349 (43.5)	975 (72.3)	374 (27.7)		6.88 (2.76–17.16)	<0.0001
Cardiovascular disorders, 48 (1.5)	40 (83.3)	8 (16.7)		4.66 (1.41–15.35)	0.011
Gastrointestinal disorders, 308 (9.9)	269 (87.3)	39 (12.7)		2.78 (1.06–7.33)	0.038
Genito–urinary disorders, 70 (2.3)	64 (91.4)	6 (8.6)		2.78 (0.80–9.68)	0.107
Neurological disorders, 277 (8.9)	264 (95.3)	13 (4.7)		1.07 (0.37–3.11)	0.897
Endocrine and metabolic disorders, 71 (2.3)	68 (95.8)	3 (4.2)		0.70 (0.13–3.74)	0.678
Surgical, 598 (19.3)	562 (94.0)	36 (6.0)		1.78 (0.67–4.72)	0.247
Hematological and immune disorders, 22 (0.7)	19 (86.4)	3 (13.6)		3.80 (0.82–17.56)	0.087
Allergy and toxins, 56 (1.8)	53 (94.6)	3 (5.4)		0.91 (0.20–3.98)	0.902
Minor categories, 208 (6.7)	206 (99.0)	2 (1.0)		0.23 (0.04–1.22)	0.086

OR, odds ratio; CI, confidence interval. Minor categories include musculoskeletal disorders, trauma, skin and subcutaneous disorders, mental health disorders, eye disorders, and miscellaneous. ^a^ Reference category in multivariate analysis.

**Table 5 jcm-11-07522-t005:** Subgroup analysis of diagnostic group infections (CAH).

Diagnosis	Total Cases ^a^	Hyponatremia ^a^	Percentage ^b^	*p* Value
Diarrhea	487	139	28.7	0.071
Viral infections and URTI	421	113	26.8	
Pneumonia and bronchiolitis	227	63	27.8	
UTI and pyelonephritis	108	35	32.4	
Cellulitis	41	6	14.6	
Meningitis and Encephalitis	34	7	20.6	
Miscellaneous	18	6	33.3	
Sepsis and Septicemia	8	5	62.5	
Osteomyelitis	5	0	0	
Total	1349	374	27.8	

URTI, upper respiratory tract infection; UTI, urinary tract infection. ^a^ These are absolute number of cases with CAH in each subgroup. ^b^ Percentage of the total number of cases in each subgroup.

**Table 6 jcm-11-07522-t006:** A. Incidence of hospital-acquired hyponatremia during the study period. B. Incidence of hospital-acquired hyponatremia before and after 1 Oct 2014 (isotonic intravenous fluids start use in October 2014).

A.									
**Year**	**2012**	**2013**	**2014**	**2015**	**2016**	**2017**	**2018**	**2019**	**Total**
HAH	2/202 (0.99)	7/424 (1.7)	6/464 (1.3)	3/429 (0.7)	4/399 (1.0)	1/366 (0.3)	7/385 (1.8)	12/433 (2.8)	42/3102 (1.4)
B.									
**Period**		**Before October 2014 (Hypotonic Fluids Period)**				**After October 2014 (Isotonic Fluids Period)**			**Total**
HAH		14/976 (1.4)				28/2126 (1.3)			42/3102 (1.4)

HAH, hospital-acquired hyponatremia. A. describes the cases of hospital-acquired hyponatremia with respect to the total number of plasma sodium determined annually in the eight years of the study. Absolute numbers and their percentage in parentheses are used. B. describes the cases of hospital-acquired hyponatremia with respect to the total number of plasma sodium determined in each of the two periods studied. Absolute numbers and their percentage in parentheses are used. The transition in the use of maintenance intravenous fluid therapy from hypotonic to isotonic occurred progressively from June 2014 to October 2014.

## Data Availability

The dataset and source code generated during and/or analyzed during the current study are available from the corresponding authors on reasonable request.

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
