# Peer review of "Incidence and Severity of Community- and Hospital-Acquired Hyponatremia in Pediatrics"

_jcm, 2022, doi:10.3390/jcm11247522_

Round 1

Reviewer 1 Report

The authors aimed to better define the incidence of community and hospital-acquired hyponatremia and their associated risk factors. The topic is important and the authors’ intent is commendable. However, I have outlined certain concerns and feedback below. Thank you for the opportunity to review this work.

P6 L 263: The authors describe year not being a significant variable in their model and cite Table 5. I do not see a Table 5, only a Table 4 which does not include year as an analyzed variable. The authors suggest that isotonic fluids did not reduce the incidence of HAH in their cohort because hyponatremia did not decrease following a 2014 switch to isotonic fluids. However, the authors do not provide any data regarding to what degree these patients were on IV fluids, in contrast to Hoorn et al, nor do we know how universally compliant house staff were with following the switch to isotonic fluids for their patients.

P7 L 270: “transfer of critically ill children to tertiary intensive care unit after initial stabilization” – This sentence was confusing for me, can the authors confirm, this study did not include any patients going to or from the ICU, that ICU-level patients were transferred to another center? The authors state that this lack of ICU patients may have decreased the HAH incidence but wouldn’t it have also reduced their reported CAH incidence, that critically ill patients who may have also had CAH would not have been seen at the authors’ institution?

P3 L 113: Can you clarify if readmissions who had sodiums drawn were present and if so, if they were excluded from analysis or classified as CAH vs HAH.

Table 4. I’m confused by the rationale for including LOS as a variable in your community-acquired hyponatremia multivariable model. How would LOS increase the risk for hyponatremia that was already acquired prior to arriving at the hospital (community-acquired)? If anything, LOS should be thought of as a secondary outcome, with hyponatremia, along with other known contributors to prolonged LOS, entered into a separate multivariable model. Further, cutting up LOS into arbitrary cutoffs is not the best practice statistically speaking. I would consider keeping it as a continuous variable/outcome.

I think there should be more analysis and discussion regarding the diagnostic groups’ contribution towards hyponatremia risk. Bronchiolitis, pyelonephritis, heart failure, and GI infection have all been linked to hyponatremia. A discussion of these diagnosis groups in the context of each other would be enlightening. I would also suggest a subgroup analysis on infection, breaking it down perhaps by system, looking at which specific organ system infections are most associated with CAH. Also, can the authors provide a rationale for making respiratory disorders the referent group?

Table 4. By definition, your referent groups should not have P values.

Author Response

Reviewer 1.
The authors aimed to better define the incidence of community and hospital-acquired hyponatremia and their associated risk factors. The topic is important and the authors’ intent is commendable. However, I have outlined certain concerns and feedback below. Thank you for the opportunity to review this work.

We greatly appreciate the reviewer's comments, suggestions, and corrections. Undoubtedly, with the changes made after its corrections, the manuscript has gained invalidity and its scientific quality has increased.

P6 L 263: The authors describe year not being a significant variable in their model and cite Table 5. I do not see a Table 5, only a Table 4 which does not include year as an analyzed variable. The authors suggest that isotonic fluids did not reduce the incidence of HAH in their cohort because hyponatremia did not decrease following a 2014 switch to isotonic fluids. However, the authors do not provide any data regarding to what degree these patients were on IV fluids, in contrast to Hoorn et al, nor do we know how universally compliant house staff were with following the switch to isotonic fluids for their patients.

We apologize to the reviewer for this error on our part since table 5 has not been included, by oversight. We have included it (It is referred now as table 6) and clarified the issue in the text. Prior to Oct-14, all patients were prescribed hypotonic intravenous fluids (saline 0.33%+ D5%) with total volumes per day according to Holliday & Segar formula. Change in the use of hypotonic fluids for isotonic fluids was done in the hospital pharmacy from October 2014 as the hypotonic fluids were not available for use. Thus, from October 2014 all the intravenous fluids for paediatric patients were isotonic. Unfortunately, the data for details of type of fluids, rate and the indication for intravenous fluids is not included in the coding data collection. There is also limitation of further exploring the medical records in select patients due to the current Spanish legal regulation. This has been a major limitation of the retrospective nature of the study. Thus, ideally a prospective study with all the data points in consideration with an appropriate ethical approval might be able to provide some valuable information. L 318.

P7 L 270: “transfer of critically ill children to tertiary intensive care unit after initial stabilization” –This sentence was confusing for me, can the authors confirm, this study did not include any patients going to or from the ICU, that ICU-level patients were transferred to another center?

We apologize for the confusion in the statement. Our hospital does not have a PICU. If a paediatric patient is admitted to our hospital and if the clinical condition requires admission to an intensive care unit, he/she is stabilized and transferred to one of our two reference tertiary hospitals in the cities of Toledo or Albacete.

These patients firstly in our hospital, are stabilized in the emergency department or if they have deteriorated on the paediatric ward, then in the high dependency unit on ward, and then transferred by a transport team to the tertiary centre where a PICU bed is available.

The authors state that this lack of ICU patients may have decreased the HAH incidence but wouldn’t it have also reduced their reported CAH incidence, that critically ill patients who may have also had CAH would not have been seen at the authors’ institution?

Children presenting to the emergency department who are critically ill would have a blood sample collected for a probable diagnosis and initial management for stabilization. These patients are then transferred for further management to a PICU, mostly within 2-4 hours. Hence if these children had hyponatremia at presentation, it would be CAH and would be captured in our data.

P3 L 113: Can you clarify if readmissions who had sodiums drawn were present and if so, if they were excluded from analysis or classified as CAH vs HAH.

We appreciate your comment. We have reviewed the dates of admission of those patients who had more than one admission during the study period and none of them implied a readmission because they were different pathologies at distant times from each other.

Table 4. I’m confused by the rationale for including LOS as a variable in your community-acquired hyponatremia multivariable model. How would LOS increase the risk for hyponatremia that was already acquired prior to arriving at the hospital (community-acquired)? If anything, LOS should be thought of as a secondary outcome, with hyponatremia, along with other known contributors to prolonged LOS, entered into a separate multivariable model. Further, cutting up LOS into arbitrary cutoffs is not the best practice statistically speaking. I would consider keeping it as a continuous variable/outcome.

Thank you very much for this valuable suggestion. We agree that the bivariate as well as multivariate analysis would not establish causality, on the contrary it would provide statistical association between the variables and in addition suggest the most significant variable/s. LOS is only associated to CAH but it doesn’t mean causality. We completely agree with that LOS should be considered as a continuous variable. We have made the necessary changes in the text and tables. We have expressed LOS as a continuous variable in table 1 and table 4 and we have repeated the process of analysis (bivariate and multivariate) and changed the results in table 4.

I think there should be more analysis and discussion regarding the diagnostic groups’ contribution towards hyponatremia risk. Bronchiolitis, pyelonephritis, heart failure, and GI infection have all been linked to hyponatremia. I would also suggest a subgroup analysis on infection, breaking it down perhaps by system, looking at which specific organ system infections are most associated with CAH.

Thank you very much for this suggestion of subgroup analysis in our largest diagnostic group, infections. We have done the subgroup analysis which is documented in Table 5. We have also included this in the discussion section. Line from 285 to 316.

Also, can the authors provide a rationale for making respiratory disorders the referent group?

We also appreciate this observation from the reviewer. We chose respiratory disorder as reference group because in the bivariate analysis “respiratory disorders” had less weight in the model with one of the lowest number of hyponatremia cases. We also performed the analysis with various probes and finally choose the model with respiratory group as a reference category in view of statistically best result. We believed that it would be easier for the readers, probably clinicians and not specialists in statistics, to understand the results with an OR > 1 (increase risk of the event, hyponatremia) than < 1 (protect from the event of hyponatremia).

Table 4. By definition, your referent groups should not have P values.

Apologies for oversight. We have removed the p value from the reference category in table 4.

Reviewer 2 Report

The presented manuscript describes the results of a retrospective study examining the incidence of hyponatremia in patients admitted to general pediatric ward of a secondary care hospital. While the general idea beyond the study is interesting and might provide valuable information for everyday handling of pediatric patients, the study and the manuscript have many flaws which needs to be addressed.

1) The study patients were divided to having either Community Acquired Hyponatremia (CAH) or Hospital Acquired Hyponatremia (HAH). It is explained that plasma sodium level of the first blood sample obtained during the care process (emergency and elective admissions) of  each patient was considered for the incidence of CAH, while any further low sodium level/s after the first normal one was considered for the incidence of HAH. Since it is written that every patient who had at least one plasma sodium measured was included, it is unclear how hyponatremia in those patients with only one measure was classified, as CAH or HAH? I believe that one measure is not enough for any of this classifications. Please comment.
2. No information or explanation has been provided on the type of intravenous fluids used in study participants, now how many of them used parenteral fluids. Since this is an important cause of iatrogen, or hospital acquired hyponatremia, I believe it is necessary to include this information as well.

3. SIADH is one of the most important cause of hypontremia in children and it is known that it can be induce by various diseases and environmental factors. SIADH as a cause of hyponatremia has not been mentioned in the manuscript, which needs to be corrected.

4. In terms of everyday clinical practice, it is not clear what is the benefit or take away message of the study? Should we be more concerned about hyponatremia in patients from two to eleven 299 years old admitted with infections, cardiovascular and gastrointestinal disorders? What can we do to prevent hyponatremia in such patents? 

5. Did you notice any changes in hyponatremia depending on the season (i.e. summer vrs winter)?

Author Response

Reviewer 2
The presented manuscript describes the results of a retrospective study examining the incidence of hyponatremia in patients admitted to general pediatric ward of a secondary care hospital. While the general idea beyond the study is interesting and might provide valuable information for everyday handling of pediatric patients, the study and the manuscript have many flaws which needs to be addressed.
We greatly appreciate this reviewer's comments, suggestions, and corrections. Undoubtedly, the manuscript has gained in validity and its scientific quality has increased with the changes made after its corrections.

  1. The study patients were divided to having either Community Acquired Hyponatremia (CAH) or Hospital Acquired Hyponatremia (HAH). It is explained that plasma sodium level of the first blood sample obtained during the care process (emergency and elective admissions) of each patient was considered for the incidence of CAH, while any further low sodium level/s after the first normal one was considered for the incidence of HAH. Since it is written that every patient who had at least one plasma sodium measured was included, it is unclear how hyponatremia in those patients with only one measure was classified, as CAH or HAH? I believe that one measure is not enough for any of this classifications. Please comment.

We fully understand the reviewer's comment and observation. Our investigation is a pragmatic, uncontrolled, observational and retrospective study that describes clinical practice as it occurs. Hence, many of the patients who are admitted have only one initial sodium determination. We have clarified this in Methods section to obtain the incidence of hyponatremia. All the cases of hyponatremia in the first determination that did not have any further sodium determination were cases of mild hyponatremia with normally mild and moderate acute processes that did or did not have intravenous maintenance fluids. All these patients improved clinically and were discharged without complications. We have attached flow diagrams of the progression of children with CAH and HAH (Figure 1 and 2). Majority of the children with CAH did not have a repeat blood sample (63.2%) suggesting there were no clinical concerns for repeat blood analysis. About two-thirds of the children with a second blood sample had normal sodium levels suggesting clinical improvement. Similarly, about half (21/42) of the children with HAH had a sodium level of 134 mEq/L with blood samples not repeated in 59% of children suggesting it to be of only biochemical significance.

We have updated this in results section (subsection 3.6), referring the readers to figures 1 and 2. Line 213.

  1. No information or explanation has been provided on the type of intravenous fluids used in study participants, now how many of them used parenteral fluids. Since this is an important cause of iatrogenic, or hospital acquired hyponatremia, I believe it is necessary to include this information as well.

Thank you very much for highlighting the most important risk factor for HAH which is intravenous fluids. The data which was collected retrospectively in our study was from the Hospital Information System (HIS). The data that is entered is based on diagnostic and procedural codes. Intravenous fluids administration is recorded only as fluids given in procedure code and no other data pertaining to fluids is collected like the type, rate and duration of fluids. These details are accessible only from the physical notes in the prescription and nursing documents which was not possible due to the data protection laws in Spain. The ethical approval is granted on the condition of that de-identification of patients is prohibited under the state law. Specific security measures were adopted to prevent re-identification and access by unauthorised third parties. This has been a major limitation of the retrospective nature of the study. Thus, ideally a prospective study with all the data points in consideration with an appropriate ethical approval might be able to provide some valuable information. We have clarified in our limitations in discussion section (Line 152 and line 318).
We analysed the impact of hypotonic vs isotonic fluids on the incidence of HAH by reviewing the data following a change in the prescription policy of only isotonic fluids after October 2014. Prior to Oct-14, all patients were prescribed hypotonic intravenous fluids (saline 0.33%+ D5%) with total volumes per day according to Holliday & Segar formula. Thus, from October 2014 pharmacy dispensed all the intravenous fluids for paediatric patients as isotonic fluids. This change did not affect the yearly incidence of HAH which has been shown in the now submitted Table 6 of the manuscript. Due to technical error this table was not included in the previous manuscript for which we apologize for the error.

  1. SIADH is one of the most important cause of hypontremia in children and it is known that it can be induce by various diseases and environmental factors. SIADH as a cause of hyponatremia has not been mentioned in the manuscript, which needs to be corrected.

We thank the reviewer for this suggestion. We have included this in the discussion with sub-group analysis of infections. Line 285, 303 and 307.

  1. In terms of everyday clinical practice, it is not clear what is the benefit or take away message of the study? Should we be more concerned about hyponatremia in patients from two to eleven years old admitted with infections, cardiovascular and gastrointestinal disorders? What can we do to prevent hyponatremia in such patents?
    Many thanks for the opportunity for us to clarify this point. We would like to suggest that these children are at more risk of hyponatremia and would need closer monitoring especially when hyponatremia is noted with infections, cardiovascular and gastrointestinal disorders. Our study has limitations of retrospective nature and non-availability of any clinical data (early warning scores or other laboratory parameters like CRP) for further analysis. We have attempted to explain the possible importance of hyponatremia in infection with the sub-analysis (In discussion section). We would recommend further research in the outcome of children in this age group and hyponatremia with a well-designed prospective study collecting these data points (in discussion section). Most of the children presented to the emergency department with hyponatremia, suggesting more research in concept of CAH needs to be directed exploring community practices of management of infection or other diseases.

  2. Did you notice any changes in hyponatremia depending on the season (i.e. summer vrs winter)?
    Thank you for the commentary. Our Table 4 reflected this analysis. There was lower percentage of hyponatremia in summer as compared to winter and rest of the year. Although this was statistically significant in bivariate analysis it did not contribute to the multivariate analysis.
    We have updated this in the manuscript line 189.

Round 2

Reviewer 1 Report

I appreciate the authors’ prompt response. I have one remaining issue outlined below, thank you.

Table 4: Please remove LOS from your model. The whole purpose of a multivariable model is to identify variables that impact a dependent variable/outcome. Multivariable models were created to predict. This is why in a multivariable model the outcome is called "dependent". The outcome is explicitly defined as being dependent on the variables entered into the model. In other words, the purpose of multivariable logistic regression is to find variables that forecast an outcome. LOS cannot forecast CAH because it follows after the CAH has already occurred. Therefore, your model is producing reverse causation error.

Regarding multivariable models, the TRIPOD guidelines mandate that authors use separate training and validation datasets if they want to call an independent variable a “predictor” of an outcome. As a result, many authors justifiably use the word “associated” because they have not done the step of using training and validation data with their multivariable model. But these authors still follow the core principle of using multivariable models to identify variables that forecast an outcome.

Therefore, by saying CAH is "associated" with prolonged LOS (P 8 Line 357), you are co-opting language that implies causality/prediction. In order to properly use a multivariable model and to truly say associated without producing reverse causation error, you have to build a multivariable model with LOS as the outcome (dependent variable) and with CAH as one of the candidate independent variables, along with other variables that have been shown to predict for LOS. An alternative is to do an unadjusted regression analysis just between CAH and LOS, with LOS as the outcome if a multivariable model is beyond scope.

Author Response

We appreciate this precise and concise explanation from the reviewer. Thanks to that we have clarified some ideas in our study.

We have removed LOS from the analysis. We repeated the analysis and also made appropriate changes in the manuscript (text and table 4) as suggested by the reviewer. 

We also deleted reference number 26 (Pham SL et al) pertaining to LOS.

Reviewer 2 Report

The authors have successfully addressed all of the concerns raised by rewivers. 

Author Response

We appreciate the reviewer's comments